# Prevalence of loneliness amongst older people in high-income countries: A systematic review and meta-analysis

Kavita Chawla[☉], Tafadzwa Patience Kunonga[iD]\*[☉], Daniel Stow, Robert Barker[iD],
Dawn Craig, Barbara Hanratty

Population and Health Sciences Institute, Newcastle University, Newcastle upon Tyne, United Kingdom

[☉] These authors contributed equally to this work.
\* patience.kunonga@ncl.ac.uk

**Data Availability Statement:** All relevant data are within the paper and its Supporting Information files.

## Abstract

### Background and objective

Loneliness is associated with increased rates of morbidity and mortality, and is a growing public health concern in later life. This study aimed to produce an evidence-based estimate of the prevalence of loneliness amongst older people (aged 60 years and above).

### Study design and setting

Systematic review and random-effects meta-analysis of observational studies from high income countries 2008 to 2020, identified from searches of five electronic databases (Medline, EMBASE, PsychINFO, CINAHL, Proquest Social Sciences Premium Collection). Studies were included if they measured loneliness in an unselected population.

### Results

Thirty-nine studies reported data on 120,000 older people from 29 countries. Thirty-one studies were suitable for meta-analysis. The pooled prevalence estimate of loneliness was 28.5% (95%CI: 23.9% - 33.2%). In twenty-nine studies reporting loneliness severity, the pooled prevalence was 25.9% (95%CI: 21.6% - 30.3%) for moderate loneliness and 7.9% (95%CI: 4.8% - 11.6%) for severe loneliness ($z$ = -6.1, $p$ < 0.001). Similar pooled prevalence estimates were observed for people aged 65–75 years (27.6%, 95%CI: 22.6% - 33.0%) and over 75 years (31.3%, 95%CI: 21.0% - 42.7%, z = 0.64, p = 0.52). Lower levels of loneliness were reported in studies from Northern Europe compared to South and Eastern Europe.

### Conclusions

Loneliness is common amongst older adults affecting approximately one in four in high income countries. There is no evidence of an increase in the prevalence of loneliness with age in the older population. The burden of loneliness is an important public health and social problem, despite severe loneliness being uncommon.

**Funding:** This work was supported by the National Institute for Health Research (NIHR) School for Primary Care Research (SPCR, 2016DS to D.S). The views expressed are those of the authors and not necessarily those of the NHS, the NIHR or the Department of Health. The funders had no role in study design, data collection and analysis, decision to publish, or preparation of the manuscript.

**Competing interests:** The authors have declared that no competing interests exist.

## PROSPERO registration

CRD42017060472.

## Introduction

Loneliness is acknowledged as a public health concern, particularly in later life [1]. It is has been defined as an undesirable subjective experience, arising from unfulfilled intimate and social needs [2]. Loneliness is accepted to have a negative impact on quality of life and wellbeing [3] and the overall risk to health from loneliness has been compared to that of smoking and obesity [4]. Evidence is growing for an association with a wide range of adverse mental [5] and physical health outcomes, including dementia [6], long term conditions such as frailty [7], cardiovascular disease [8] and premature mortality [9]. Recent studies have also suggested that lonely older people are more likely to be living with multiple health conditions [10], and that loneliness may be linked with increased use of some health and social care services [11].

Tackling loneliness is already a policy imperative in many countries, and England and Wales introduced a dedicated strategy in 2018 [12]. Yet despite this, our understanding of the size of the public health problem is limited [13, 14]. To date, reviews of the evidence have been unsystematic [15], selective [16, 17], or rapid in their approaches [18]. They report prevalence figures of between 2% and 34% for older people with a suggestion that chronic loneliness may affect 10 to 15% of people at all ages [18]. Studies from the UK, New Zealand and elsewhere point to a U-shaped distribution in the prevalence of loneliness across the life-course, but with little agreement over the age at which peaks and troughs in loneliness may be observed [19–21]. There is a perception that loneliness is becoming more common over time, with possible causes being changes in living arrangements, a rise in solitary living [22], smaller families [22], the loss of community resources such as libraries and post offices and the advent of the internet [23]. However, analyses of repeated cross-sectional studies in the UK (1946 to1999) [24], and Sweden (1994 to 2014) [25] suggest that the prevalence of loneliness has remained stable over time.

A sound knowledge of the epidemiology of loneliness is essential to support intervention design and development, and service planning. This review aims to determine the prevalence of loneliness amongst the general population of older people in high income countries. Where data are available, it will also seek to identify any differences in the prevalence of loneliness by age amongst older people.

## Materials and methods

This review adopted methods recommended by the Centre for Reviews and Dissemination [26], reporting follows PRISMA guidance [27]. A protocol was registered prospectively with PROSPERO (CRD42017060472).

### Search strategy

Five electronic databases were searched from 2008 to July 2020 (Medline, Embase, PsychINFO, CINAHL, Social Sciences Premium Collection), using a search strategy developed and tailored for each database with an information scientist (data in S1 File). Thesaurus headings and key words related to loneliness and older people were used. Grey literature (The Campaign to End Loneliness, Gulbenkian Foundation, and AGE UK websites), ageing journals and reference

lists of included studies were also searched. No date or country restrictions were applied at this stage, but we restricted the search to English language material.

## Inclusion criteria

Studies were included if they provided cross-sectional data on any measure of loneliness within a defined population of adults from high income countries, defined by the World Bank [28]; published between 2008 and 2020 aged 60 years and above. Studies focusing on institutionalised populations only were excluded, however because we were looking at the prevalence amongst older people, studies with a small proportion of people in care homes were included. Highly selected samples, such as groups of patients with a single disease (for example heart disease), specific characteristics, or role (e.g., caregivers) were excluded.

## Screening and data extraction

The selection process consisted of two stages of screening, conducted by two reviewers. Articles were exported from Endnote X9 to Rayyan, an online bibliographic database [29]. The inclusion criteria were tested and refined on sample of titles and abstracts to ensure that they were robust enough to capture relevant articles. One researcher then screened all titles and abstracts of the included articles and another researcher checked 10% of these for accuracy. All articles included for full-text examination were independently checked by both reviewers. Disagreements between the reviewers were resolved either by discussion between the reviewers or with arbitration from another member of the review team.

Data were extracted into a bespoke form, developed using relevant guidance [30]. In addition to standard study and population demographic data, information was extracted on the measurement tools/scales, cut-offs used to define loneliness and frequency or intensity of loneliness. Authors were contacted for further information, where necessary, and one response received.

## Classification of loneliness

Loneliness was defined by any self-reported measure of loneliness, ranging from a single items question to a multi-item tool [31]. There are numerous tools available to assess loneliness, and if studies measured the number or proportion of people who were experiencing loneliness, irrespective of the loneliness measure used, the study was included.

The experience of loneliness can vary in intensity and frequency within the same individual during their lifetime and therefore a point prevalence may be difficult to interpret. It is believed that both situational and chronic loneliness are associated with adverse health conditions [32]. Therefore, this review sought to measure the prevalence of any feelings of loneliness. For this study we categorised people as lonely if they answered positively to feelings of loneliness such as feeling lonely 'often' or 'always' or 'sometimes.' If they gave a negatively worded response such as 'not' lonely or 'not very' lonely this was categorised as not lonely.

For those studies which provided the number of participants reporting different intensities or frequencies of loneliness, participants reporting the highest category of loneliness possible in the study were considered 'severely' lonely in this review.

## Risk of bias

Risk of bias was assessed using the standardised the Joanna Briggs Checklist for Studies Reporting Prevalence Data [33]; 30% of studies were assessed by two researchers. In this

review, risk of bias was judged to be low, moderate, or high if ≤2, 3–4, or 5+ criteria respectively were not fulfilled. However, no studies were excluded because of risk scores.

## Meta-analysis

Where suitable statistical data were reported, we conducted a pooled meta-analysis using Metafor [34] in R Core Team [35]. Heterogeneity was assessed with the $I^2$ statistic, with a value over 50% taken to indicate substantial heterogeneity [36]. Random-effects meta-analysis was conducted with pre-specified subgroups to explore potential sources of heterogeneity. Subgroup comparisons were made using a Wald type estimator [37]. Data were transformed using the Freeman-Tukey double arcsine transformation [38].

## Results

Thirty-nine studies were included in this review, after screening of 12,849 titles/abstracts and review of 139 full texts. Reasons for exclusion included: not high income country, having a disease or role-related to those people with a specific health condition (for example heart disease), aged <60 years, baseline not a general population, data not reported, a study reported in more than one paper, published before 2008, and loneliness score were divided into quartile and quintiles and used as cut-offs. The study selection process is shown in Fig 1; details and main findings of the included studies in Table 1.

The included studies provided information on approximately 140,000 older people (age 60 years and over) from 30 countries. Most of the research was conducted in European countries [7, 25, 39–59, 72, 73] with four from the USA [64–67], two from each of Israel [60, 61] and New Zealand [19, 62] and single studies from Singapore [68] and Australia [63]. Four studies provided estimates of prevalence across a large proportion of Europe [69–71, 73]. Living circumstances were reported in seven studies; people living in institutions were included in four studies [25, 39, 49, 50], excluded in three [46, 52, 61] and no data available from the remainder.

All thirty-nine studies were cross-sectional in design. Data were collected by face-to-face interview [7, 19, 25, 39, 41, 42, 46, 47, 49–55, 57–61, 64, 67–71], postal questionnaires [40, 43–45, 48, 56, 62, 63, 73] and online questionnaires [65, 66, 72].

## Measurement of loneliness

Loneliness was measured using either a single item question, or one of two well established scales. Eight studies used the De Jong Gierveld scale [49, 50, 54, 56, 62, 68, 73], and all but one study [70] employed the longer 11-point version. The UCLA loneliness measure was used in five studies [7, 65, 66, 68, 72]. Respondents were asked to consider experiences of loneliness in the previous week [63, 64, 69, 71] four weeks [19, 61] or year [59]. In eleven studies, no time frame was specified [25, 39–41, 43, 44, 46, 47, 50, 51, 60]. One study used an 11-tem Loneliness Scale designed for large surveys [67].

## Risk of bias

Using the Joanna Briggs Checklist for Studies Reporting Prevalence Data, the majority (28 out of 39) of studies were judged to have a low risk of bias [7, 25, 39–45, 50–53, 55–59, 61, 64, 66, 67, 69–73]. Ten studies were judged to be moderate risk [19, 46–48, 54, 60, 62, 63, 65, 68] and only one study based in the UK was judged to be at high risk of bias [49]. The potential for bias was identified in 4 out of 9 domains, mainly due to not having enough participants to address the target population. The domains with the greatest risk of bias relate to coverage bias

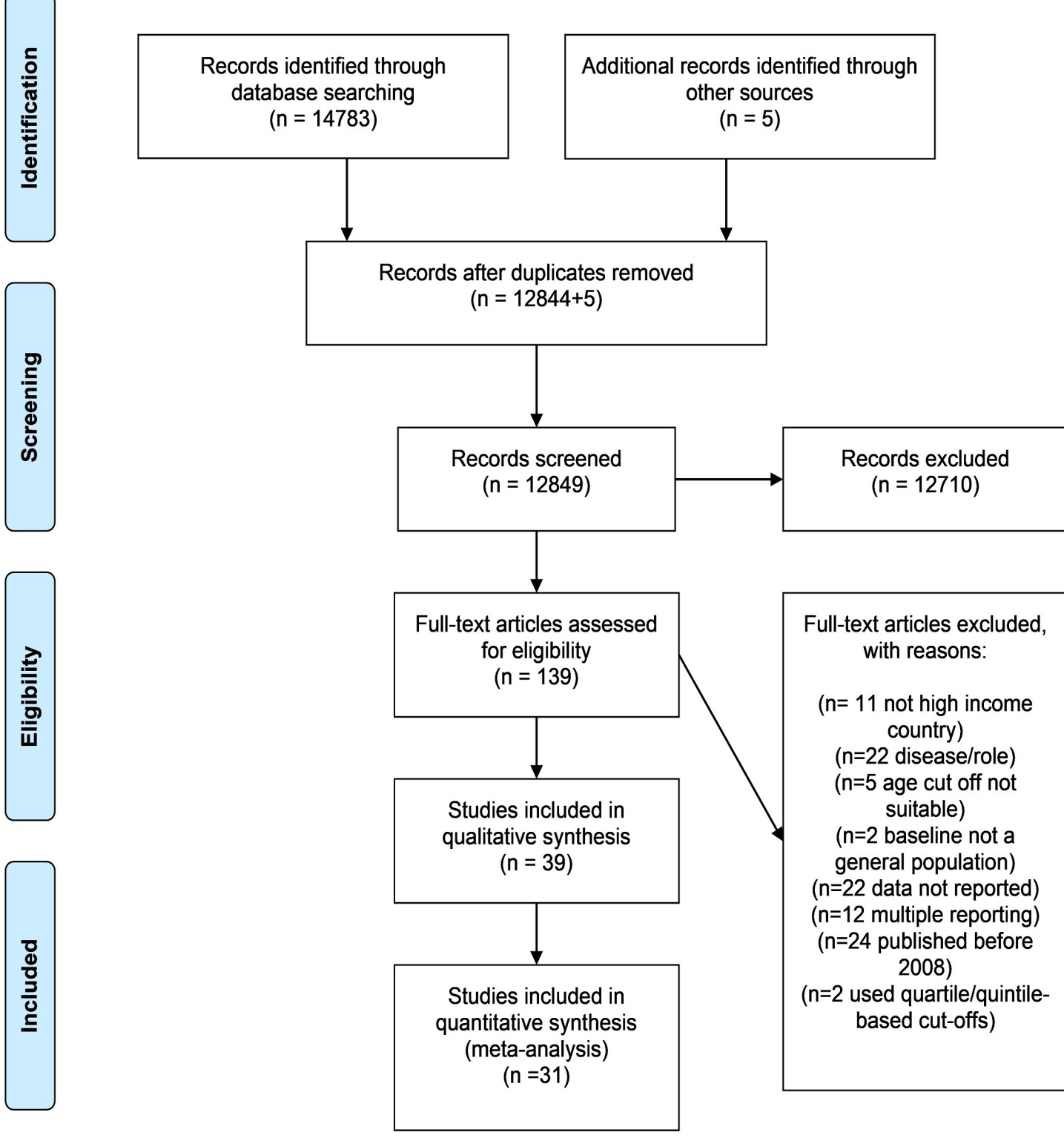

**Fig 1. Flow diagram of included studies.**

**Table 1. Summary of details and findings of studies included in the review.**

| Study Authors | Location | National/sub-national | Year of Collection | Age | Question asked or tool used. (responses coded as 'lonely' in this review) | Proportion lonely % (actual proportion, if available) |
|---|---|---|---|---|---|---|
| Dahlberg et al. [25] | Sweden | National-SWEOLD study | 1992–2011 | 77+ | Do you ever feel lonely? (always or frequently) | 13.1% (336/2572) |
| Nyqvist et al. [39] | Sweden | Sub national- region of Northern Sweden | 2002–2012 | 85+ | Do you ever feel lonely? (sometimes or often) | 49.7% (514/1034) |
| Djuukanovic et al. [40] | Sweden | National-authors own sample | 2010 | 65–80 | Do you ever feel lonely? (sometimes or often) | 27.55 (1833/ 6659) |
| Aartsen & Jylha [41] | Finland | Sub national-Tampere | 1979 | 60–89 | Do you feel lonely? (sometimes or often) | 29.2% (135/463) |
| Heikkinen & Kauppinen [42] | Finland | Sub national-Jyvaskyla | 1988 | 65+ | I am: (very lonely or rather lonely or lonely now and then) | 25,2% (158/628) |
| Eloranta et al. [43] | Finland | Sub national-Turku | 1991 | 70+ | Do you suffer from loneliness? (always or often or sometimes) | 22.3% (439/1966) |
| Tilvis et al. [44] | Finland | National | 2002 | 75+ | Do you suffer from loneliness? (sometimes or often or always) | 46.1% (1781/3858) |
| Lasgaard et al. [45] | Denmark | Sub national- Central Denmark Region | 2013 | 60+ | Three Item Loneliness scale- 7 = severe loneliness | 12.2% (1454/11961) |
| | | | | 60+ | 5–6 = moderate Loneliness | |
| Bergland et al. [46] | Norway | Oslo | 1997–1998 | 75+ | Do you find yourself lonely? (Quite often, sometimes) | 33.2% (102/307) |
| | | | | | | Females only |
| Nicolaisen & Thorsen [47] | Norway | National | 2002 | 60–80 | Do you feel lonely? (sometimes or often) | 30.% (415/1378) |
| Tomstad et al. [48] | Norway | Southern Norway | 2010 | 65+ | Do you often feel lonely? (Yes) | 11.6% (239/2052) |
| Scharf & de Jong. [49] UK sample | England | 3 deprived urban areas | 2000/2001 | 60+ | De Jong Gierveld | 56% (280/500) |
| | | | | | 3 or more out of 11 | |
| Brittain et al. [50] | England | Newcastle- Newcastle 85+ study | 2006 | 85 years | Rate your loneliness ('always', 'often', 'sometimes') | 43% (325/750) |
| Gale et al. [7] | England | English Longitudinal Study of Ageing (ELSA) | 2008/9 | 60+ | Revised UCLA loneliness scale | 44.1% (1034/2346) |
| | | | 2010/11 | 60+ | | 45.2% (1272/2817) |
| Victor & Bowling [51] | United Kingdom | National | 1999–2000 | 65+ | I am: (always or often or sometimes lonely) | 38.0% (379/997) |
| Dahlberg & Mckee [52] | United Kingdom | Barnsley- authors own sampling | 2008 | 65+ | De Jong Gierveld | 46% (563/1224) |
| | | | | | 3 or more out of 11 | |
| Thomas [53] | United Kingdom | Uk- ONS 2014 Opinions and Lifestyle | 2014/15 | 65+80+ | 'On a scale where 0 is not at all lonely and 10 is extremely lonely, how lonely are you?' | 14.5% in 65–79-year-olds |
| | | | | | | 29.2% in 80+ |
| | | | | | Highly lonely (6–10) | |
| Julsing et al. [54] | Netherlands | Zutphen | 1985 | 65–85 | De Jong Gierveld Scale | 42% (302/719) |
| | | | | | 4 or more out of 11 | Males only |
| Holwerda et al. [55] | Netherlands | Amsterdam | 1991 | 65–84 | Do you feel lonely? | 21.5% (859/4004) |
| Scharf & de Jong [49] Netherlands sample | Netherlands | 3 regions-NESTOR survey | 1992 | 60+ | De Jong Gierveld Scale | 38% (1333/3508) |
| | | | | | 3 or more out of 11 | |
| Honigh-de Vlaming et al. [56] | Netherlands | Gelderland | 2005/2010 | 65+ | De Jong Gierveld Scale | 39% (3761/9641) |
| | | | | | 3 or more out of 11 | |
| Tabue Teguo et al. [57] | France | 2 districts | 1989 | 65+ | 'I felt lonely'- | 13.8% (498/3620) |
| | | | | | Sometimes 1–2 days, moderately 3–4 days, most of the time 5–7 days of the week | |
| Golden et al. [58] | Republic of Ireland | Dublin | 1993–2002 | 65+ | Did you feel lonely last month? (yes) | 34.8% (452/1299) |

*(Continued)*

**Table 1.** (Continued)

| Study Authors | Location | National/sub-national | Year of Collection | Age | Question asked or tool used. (responses coded as 'lonely' in this review) | Proportion lonely % (actual proportion, if available) |
|---|---|---|---|---|---|---|
| Molloy et al. [59] | Republic of Ireland and Northern Ireland | National | Not stated | 65+ | How often in the last 12 months have you been bothered by loneliness? (very often or quite often) | 15.0% (305/2033) |
| Stessman et al. [60] | Israel | Jerusalem | 1990 | 70 | How often are you lonely? (often/v often) | 17.9% (108/60) |
| Cohen-Mansfield et al. [61] | Israel | National | 1989–1992 | 75+ | How often did you feel lonely during the last month? (sometimes, mostly, almost every day) | 38.1% (437/1147) |
| La Grow et al. [62] | New Zealand | National | Not stated | 65+ | De Jong Gierveld loneliness scale 3 or more out of 11 | 52.4% (174/332) |
| Ministry of Social Development [19] | New Zealand | National | 2014 | 65–74 years | Did you feel lonely over the last four weeks? | 9.6% lonely in 65–74-year-olds |
| | | | | 75+ | (All or most or some of the time) | 12.5% lonely in over 75s |
| Franklin & Tranter [63] | Australia | National | 2007 | 65+ | Do you often experience loneliness? | 20.5% |
| | | | | | Loneliness is a serious problem for me. | 15.0% |
| Theeke [64] | USA | National Health and Retirement Study | 2002 | 65+ | Have you been feeling lonely for much of the last week? (yes) | 19.3% (1727/8932) |
| AARP [65] (American Association of Retired Persons) | USA | National | 2010 | 60–69 | UCLA loneliness scale (44 or more out of 80) | 32% in 60–69-year-olds |
| | | | | 70+ | | 25% in over 70s |
| Hawkley et al. [66] | USA | Sub-national | 2005/6 | 65+ | Three item UCLA Loneliness Scale | 28% (841/3005) |
| | | | 2015/16 | | | 31% (1480/4777) |
| Pavela et al. [67] | USA | Sub-national | 2008–2010 | 65+ | 11-item Loneliness Scale | Not reported |
| Lim &Chan [68] | Singapore | National | 2009 and 2011 | 60+ | Three item UCLA scale Scores more than never on each item | 55% (1514/2728) |
| Yang & Victor [69] | Europe | National | 2006/7 | 60+ | How much of the last week have you felt lonely? (most, all or almost all) | 10.5% (1079/10282) |
| Hansen & Slagsvold [70] | Europe | National- gender and generations survey | 2004–2011 | 60–80 | De Jong Gierveld Loneliness Scale Six or more out of 12 | 15.4% (3442/ 22429) |
| Sundstrom et al. [71] | Europe | National | 2004 | 65+ | How often have you experienced the feeling of loneliness over the past week (almost all, most, some) | 41% (3614/8787) |
| Lara et al. [72] | Spain | Sub-national | 2014–2015 | 65+ | Three item UCLA Loneliness Scale | 12.4% (210/1691) |
| Tan et al. [73] | UK, Greece, Croatia, Netherlands and Spain | National | 2015–2017 | 75+ | 6-item De Jong Gierveld Loneliness Scale | 46.4% (1007/2169) |

(domain 5) and response rate (domain 9), as detailed in, Fig 2, principally due to the absence of sufficient information on which to make a judgement.

## Prevalence

**Gender differences in prevalence of loneliness.** Nine studies reported the prevalence of loneliness by gender. Three of the nine studies found no gender differences [62, 68]. Six studies reported that females were more likely than males to be lonely. This was attributed to gender differences in the distribution of risk factors such as living arrangements, marital status, and self-rated health [6, 41, 43, 45, 50, 61].

Two articles reported on loneliness within single sex populations [46, 54]. A study in Oslo with 307 females aged over 75 years and living at home, reported that one third of the female

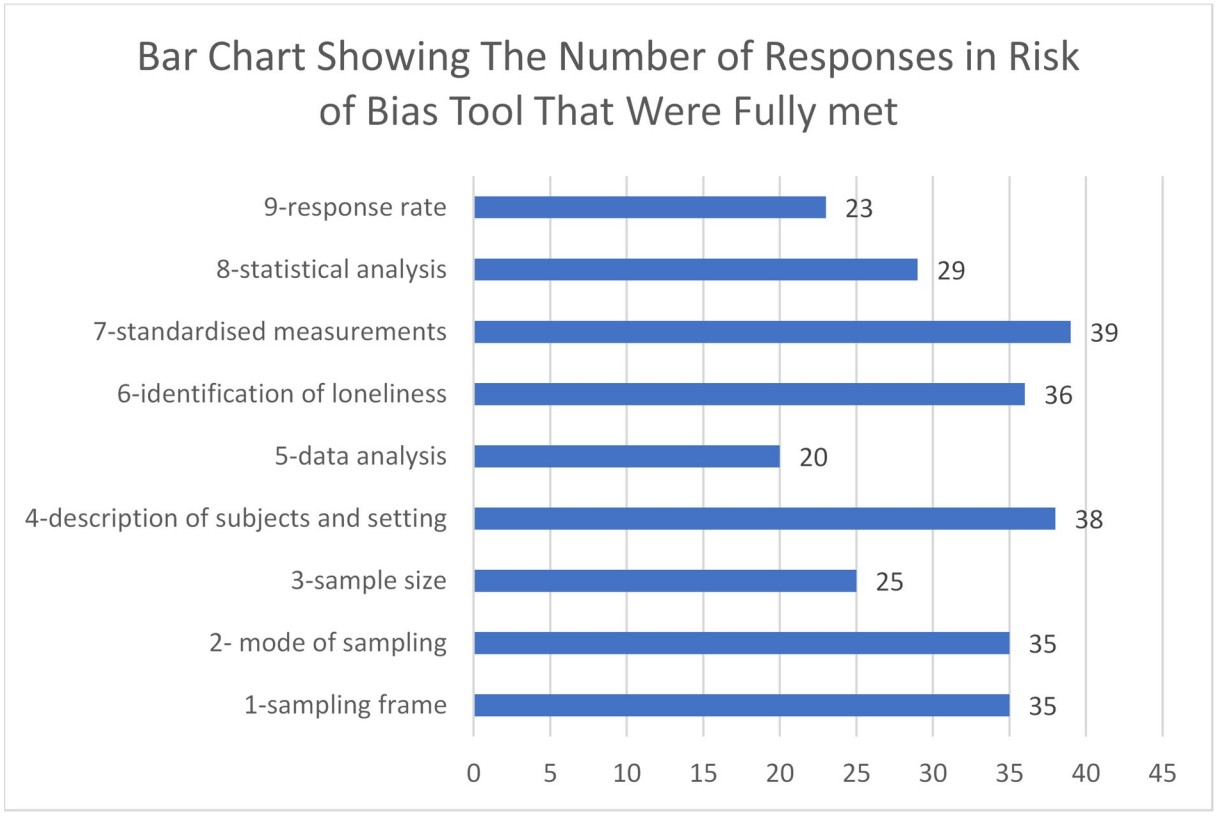

**Fig 2. Results of risk of bias.**

participants were lonely at least some of the time [46]. In Amsterdam, Julsing and colleagues' [54] study with men aged 64–84 years found a prevalence of loneliness of 43%.

**Geographical variation in prevalence of loneliness.** Although this study was not designed to investigate geographical differences, we did observe variation in the reported prevalence of loneliness within and between countries. Figures in European countries ranged from 11.6% to 56.0%. North American studies reported 19.3% to 32.0%; Israel 17.9% to 38.1%, New Zealand 9.6% to 52%, and Singapore 55%. Some of this variation may be due to the differences in the cut off points for loneliness measurements, data collection methods and sample sizes. International comparative studies describe consistent patterns, with reported prevalence higher in Mediterranean countries (e.g., Italy and Portugal) compared to northern European countries (UK, Ireland, Scandinavia) [74], but not those of the former Soviet Union [75]. Further research to better understand these geographical differences might support learning or understanding from countries with lower prevalence.

## Meta-analysis

Thirty-one of the thirty-nine studies provided data suitable for pooling (Fig 3). The pooled estimate of the prevalence of loneliness amongst people aged over 65 years in high income countries, was 28.5% (95%CI: 23.9% to 33.2%). There was a substantial degree of heterogeneity ($I^2$ 99.7%, Q-test $p < 0.001$), likely due to the differences in the types of measurement tools used, different methods of obtaining data (online, face to face, postal questionnaire), differences in response rates as well as gender.

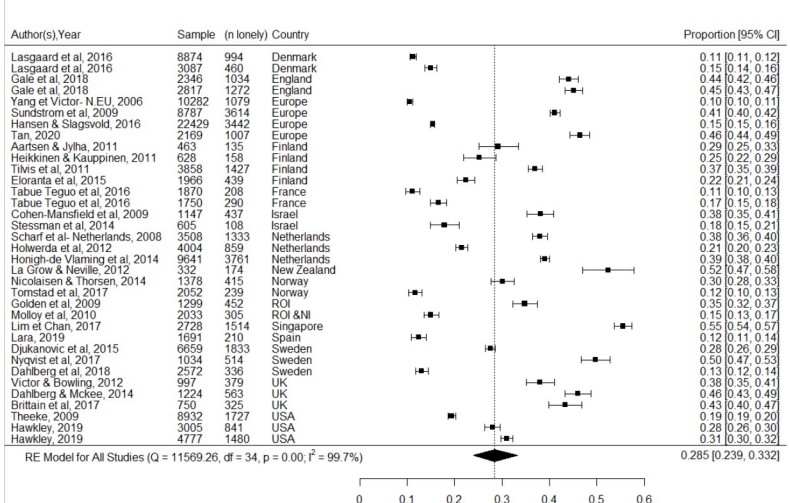

**Fig 3. Forest plot of loneliness prevalence estimates from 31 included studies.**

Two additional a priori specified subgroup analyses were conducted. The first compared people reporting severe loneliness to those reporting moderate loneliness. Twenty-nine studies provided information on loneliness severity (moderate vs severe). Pooled estimates were 7.9% (95%CI: 4.8% to 11.6%) for people reporting severe loneliness, and 25.9% (95%CI: 21.6% to 30.3%) for moderate loneliness ($z = -6.1$, $p < 0.001$ for the comparison).

The point estimate for the pooled estimate of loneliness prevalence in people aged over 75 (31.3%, 95%CI: 21.0% to 42.7%) was higher than for people age 65–75 (27.6%, 95%CI: 22.6% to 33.0%), but the confidence intervals for these estimates were wide and overlapping and the comparison was not conventionally statistically significant ($z = 0.64$, p = 0.52).

## Discussion

To our knowledge, this is the first systematic review to present pooled estimates of loneliness amongst older adults living in high income countries. Our pooled estimate of loneliness prevalence (28.5%, 95%CI: 23.9% to 33.2%) suggests that approximately 1 in 4 older adults over 60 experience some degree of loneliness at least some of the time. Our a-priori subgroup analyses suggest that a smaller proportion of people (around 1 in 12) experience severe loneliness than moderate loneliness (approximately 1 in 4). There was no suggestion that loneliness is more common in people age over 75 than in those age 65–75. However, there was some evidence that the prevalence of loneliness is lowest in northern European countries and higher in Mediterranean countries and Eastern Europe.

### Comparison with other work

The prevalence estimates from our meta-analysis sit within the previously published ranges, which are wide. There is evidence to support the fact that loneliness levels have been static over the past 70 years [76]. A study conducting a comparative analysis of four surveys from the UK between 1945 and 1999, found no change in variations of loneliness between cohorts [24]. Between five and nine percent of people were often lonely, which is similar to the estimated rate of 'severe' loneliness with our findings. The range of people feeling at least 'sometimes lonely' was more variable. In one of the surveys it was 17 percent and in the other three surveys (including the more recent survey) it was between 27 and 34 percent which is in keeping with

this review's results. In addition, published prevalence rates have been most consistent for very frequent or severe loneliness amongst the older population, at between 5 and 13% [77]. The proportion of older people who reported feeling at least sometimes lonely showed more variation, at 17% to 34%. These findings held true when data from multiple countries were examined, in a review published in 1996 which reported a prevalence of severe loneliness between 2% and 16% and moderate loneliness between 7% and 42% [78]. A recent review of loneliness in care home residents estimated the mean prevalence of moderate and severe loneliness to be higher than the general older population (at 61% and 35%), but confidence intervals are wide [79].

## Strengths and limitations

This review brings together, for the first time, data from studies of the prevalence of loneliness in unselected older populations. Our approach was systematic, using validated methods. We restricted the review to studies from high-income countries to reduce some of the cultural and socioeconomic variation between study populations. However, we acknowledge that a focus on English language publications may have led us to overlook some relevant material, although we expect that the impact on overall findings would be minimal. We sought to identify differences between people in early and later old age, but data were not available to support this.

Inclusion of all measures of loneliness used in the published studies, different measurement schedules and intensity of loneliness produced a heterogeneous group of studies. Loneliness by definition is a subjective experience. Differences in social structures, ways of life, social norms and expectations are likely to impact on the prevalence of loneliness. Therefore, this review has sought to look at the prevalence of loneliness in countries at a similar stage of economic development over recent years to attempt to produce a more homogenous population group. However, we set out to produce a comprehensive review and at present, our understanding of the natural history of different experiences of loneliness (e.g., chronic versus more intense loneliness) is insufficient to justify excluding individual studies.

In using all measures of loneliness, we have pooled people experiencing loneliness over different durations (some over the last week, some over the past year), differing intensities and also differing frequencies. It is yet unclear whether these have similar natural histories. What is known is that both chronic and recent loneliness are associated with increased mortality [80] but it is not clear how the different intensities affect health.

## Implications

Overall, we have shown that loneliness is a common experience in later life, though severe loneliness affects only a small proportion of the population. As loneliness is both distressing and associated with adverse outcomes, it is a legitimate target for action. There are numerous potential risk factors for loneliness, including personal characteristics (age, gender, ethnicity), health, disability, life events (bereavement, retirement), living circumstances and social resources (single person households, social networks, presence of a confidante), and geographical area (social disadvantage, perception of crime) [81, 82]. We have reported prevalence according to a small number of factors that are not mutable. Nevertheless, they provide useful information for directing health and societal interventions to people at highest risk. Work to identify the characteristics of people affected by severe loneliness and the aetiological factors, is needed to complement any interventions to positively impact population level experiences of loneliness. Our work also emphasises the need to deepen our understanding of effective strategies to prevent or ameliorate loneliness. There is evidence that group interventions many be more effective than individual initiatives, but the range of interventions that have been tested

is narrow, with a strong focus on befriending and telephone support [83]. The appropriateness of health or social services responses to this problem is far from clear.

## Conclusions

The idea that older age is synonymous with being lonely should be challenged. Whilst one in four older people experience loneliness at some time, severe or prolonged loneliness is uncommon, not universal, and loneliness does not increase with age. Therefore, the burden of loneliness in amongst older adults is an important public health and social problem.

## Supporting information

**S1 File. Medline search strategy.**
(DOCX)

**S2 File. PRISMA checklist.**
(RTF)

## Acknowledgments

The authors wish to thank Sarah Khan and Aalya Al-Assaf for their contributions for title and abstract screening and data extraction.

## Author Contributions

**Conceptualization:** Kavita Chawla, Tafadzwa Patience Kunonga, Daniel Stow, Robert Barker, Dawn Craig, Barbara Hanratty.

**Data curation:** Kavita Chawla, Tafadzwa Patience Kunonga, Daniel Stow, Robert Barker, Dawn Craig, Barbara Hanratty.

**Formal analysis:** Kavita Chawla, Tafadzwa Patience Kunonga, Daniel Stow, Robert Barker, Dawn Craig, Barbara Hanratty.

**Funding acquisition:** Barbara Hanratty.

**Investigation:** Kavita Chawla, Tafadzwa Patience Kunonga, Daniel Stow, Robert Barker, Dawn Craig, Barbara Hanratty.

**Methodology:** Kavita Chawla, Tafadzwa Patience Kunonga, Daniel Stow, Robert Barker, Dawn Craig, Barbara Hanratty.

**Project administration:** Kavita Chawla, Tafadzwa Patience Kunonga, Dawn Craig, Barbara Hanratty.

**Resources:** Kavita Chawla, Tafadzwa Patience Kunonga, Daniel Stow, Robert Barker, Dawn Craig, Barbara Hanratty.

**Software:** Kavita Chawla, Tafadzwa Patience Kunonga, Daniel Stow, Robert Barker, Dawn Craig, Barbara Hanratty.

**Supervision:** Tafadzwa Patience Kunonga, Dawn Craig, Barbara Hanratty.

**Validation:** Kavita Chawla, Tafadzwa Patience Kunonga, Daniel Stow, Robert Barker, Dawn Craig, Barbara Hanratty.

**Visualization:** Kavita Chawla, Tafadzwa Patience Kunonga, Daniel Stow, Robert Barker, Dawn Craig, Barbara Hanratty.

**Writing – original draft:** Kavita Chawla, Tafadzwa Patience Kunonga, Barbara Hanratty.

**Writing – review & editing:** Kavita Chawla, Tafadzwa Patience Kunonga, Daniel Stow, Robert Barker, Dawn Craig, Barbara Hanratty.

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
