## [Decision Letter · Decision Letter 0]

7 Jun 2021

PONE-D-21-14347

Prevalence of Loneliness amongst Older People in High-Income Countries: A Systematic Review and Meta-analysis

PLOS ONE

Dear,

Thank you for submitting your manuscript to PLOS ONE. After careful consideration, we feel that it has merit but does not fully meet PLOS ONE’s publication criteria as it currently stands. Therefore, we invite you to submit a revised version of the manuscript that addresses the points raised during the review process.

Please improve the discussion and elaborate in details of comment as under.

We look forward to receiving your revised manuscript.

Kind regards,

Muhammad Shahzad Aslam, Ph.D.,M.Phil., Pharm-D

Academic Editor

PLOS ONE

Journal Requirements:

2. Please amend either the abstract on the online submission form (via Edit Submission) or the abstract in the manuscript so that they are identical.

4. Please include captions for ALL your Supporting Information files at the end of your manuscript, and update any in-text citations to match accordingly. Please see our Supporting Information guidelines for more information: http://journals.plos.org/plosone/s/supporting-information.

Reviewers' comments:

Reviewer's Responses to Questions

**Comments to the Author**

1. Is the manuscript technically sound, and do the data support the conclusions?

Reviewer #1: Yes

Reviewer #2: Yes

2. Has the statistical analysis been performed appropriately and rigorously? 

Reviewer #1: Yes

Reviewer #2: Yes

3. Have the authors made all data underlying the findings in their manuscript fully available?

Reviewer #1: Yes

Reviewer #2: Yes

4. Is the manuscript presented in an intelligible fashion and written in standard English?

Reviewer #1: Yes

Reviewer #2: Yes

5. Review Comments to the Author

Reviewer #1: 1. Introduction: “Tracking loneliness is …… public health problem is limited.” Any added literature support?

2. Search strategy will be confirmed. “Five electronic databases were searched from inception to July 2020”, however, Inclusion criteria mentioned, “published between 2008 and 2020”. The search strategy was not matched. It may readers misunderstanding and difficulty to qualify methodology.

3. Classification of Loneliness: Added references for definition.

4. Definition and criteria for the “High-income countries”.

5. Strengthen and limitation: Readers lack understand of this manuscript's strengths and limitations. Please add more specific paragraphs to the end of this manuscript.

Reviewer #2: 1- Please provide data identification method clearly

2-Please provide details discussion by comparison of finding of relevant studies

3-Please rewrite the conclusion. According to author "Even though older age is not synonymous with loneliness, approximately one in four older adults

experience loneliness experience loneliness. Even though severe or prolonged loneliness is less

common, the burden of loneliness in amongst older adults is an important public health and social

problem" The paragraph is unclear and does not correspond to conclusion

4-Please elaborate prisma flow diagram particularly reason of exclusion in data screening stage

5-According to author statement". There was a very high degree of heterogeneity" Please elaborate in discussion why there is high heterogeneity observed?

6-Please provide rationale of meta-analysis results and explain in discussion

7-It is recommended to draw the illustration of Geographical variation in prevalence of loneliness within and between countries mentioned inside the manuscript.

8-please provide rationale on high risk of bias in studies identified.

6. PLOS authors have the option to publish the peer review history of their article (what does this mean?). If published, this will include your full peer review and any attached files.

Reviewer #1: **Yes: **Yun Jin Kim

Reviewer #2: **Yes: **Saima Nisar

---

## [Author Response · Author response to Decision Letter 0]

15 Jun 2021

Reviewers' comments:

Review Comments to the Author

Reviewer #1: 

1. Introduction: “Tracking loneliness is …… public health problem is limited.” Any added literature support?

Our response: 

Thank you for your suggestion. We have added the following reference to support our narrative:

Cacioppo JT, Cacioppo S. The growing problem of loneliness. The Lancet. 2018 Feb 3;391(10119):426.

Gerst-Emerson K, Jayawardhana J. Loneliness as a public health issue: the impact of loneliness on health care utilization among older adults. American journal of public health. 2015 May;105(5):1013-9.

2. Search strategy will be confirmed. “Five electronic databases were searched from inception to July 2020”, however, Inclusion criteria mentioned, “published between 2008 and 2020”. The search strategy was not matched. It may readers misunderstanding and difficulty to qualify methodology.

Our response:

Thank you for your comments. The text has been amended to “Five electronic databases were searched from 2008 to July 2020”.

3. Classification of Loneliness: Added references for definition.

Our response:

Thank you for your suggestion. We have added a reference to support our statement.

Valtorta NK, Kanaan M, Gilbody S, Hanratty B. Loneliness, social isolation and social relationships: what are we measuring? A novel framework for classifying and comparing tools. BMJ Open. 2016;6(4)

4. Definition and criteria for the “High-income countries”.

Our response:

Thank you. We have included the following statement and reference in support: 

High income countries were included as defined by the World Bank. 

World Bank Group. High Income. 2021Available from: https://data.worldbank.org/country/XD

5. Strengthen and limitation: Readers lack understand of this manuscript's strengths and limitations. Please add more specific paragraphs to the end of this manuscript.

Our response:

Thank you for your suggestion. We have added the following text:

Loneliness by definition is a subjective experience. Differences in social structures, ways of life, social norms and expectations are likely to impact on the prevalence of loneliness. Therefore, this review has sought to look at the prevalence of loneliness in countries at a similar stage of economic development over recent years to attempt to produce a more homogenous population group. 

In using all measures of loneliness, we have pooled people experiencing loneliness over different durations (some over the last week, some over the past year), differing intensities and also differing frequencies. It is yet unclear whether these have similar natural histories. What is known is that both chronic and recent loneliness are associated with increased mortality but it is not clear how the different intensities affect health.

Reviewer #2: 

1- Please provide data identification method clearly

Our response:

Thank you. We have reworded the section to include:

The selection process consisted of two stages of screening, conducted by two reviewers. Articles were exported from Endnote X9 to Rayyan, an online bibliographic database [26]. The inclusion criteria were tested and refined on sample of titles and abstracts to ensure that they were robust enough to capture relevant articles. One researcher then screened all titles and abstracts of the included articles and another researcher checked 10% of these for accuracy. All articles included for full-text examination were independently checked by both reviewers. Disagreements between the reviewers were resolved either by discussion between the reviewers or with arbitration from another member of the review team.

2-Please provide details discussion by comparison of finding of relevant studies

Our response:

Thank you for your suggestion. We have added the following text:

There is evidence to support the fact that loneliness levels have been static over the past 70 years. A study conducting a comparative analysis of four surveys from the UK between 1945 and 1999, found no change in variations of loneliness between cohorts. Between five and nine percent of people were often lonely, which is similar to the estimated rate of ‘severe’ loneliness with our findings. The range of people feeling at least ‘sometimes lonely’ was more variable. In one of the surveys it was 17 percent and in the other three surveys (including the more recent survey) it was between 27 and 34 percent which is in keeping with this review’s results.

3-Please rewrite the conclusion. According to author "Even though older age is not synonymous with loneliness, approximately one in four older adults experience loneliness experience loneliness. Even though severe or prolonged loneliness is less

common, the burden of loneliness in amongst older adults is an important public health and social problem" The paragraph is unclear and does not correspond to conclusion

Our response:

Thank you for your suggestion, we have reworded our conclusion to:

The idea that older age is synonymous with being lonely should be challenged. Whilst one in four older people experience loneliness at some time, severe or prolonged loneliness is uncommon, not universal, and loneliness does not increase with age. Therefore, the burden of loneliness in amongst older adults is an important public health and social problem.

4-Please elaborate prisma flow diagram particularly reason of exclusion in data screening stage

Our response:

Thank you for your suggestion, we have added the following statement:

Reasons for exclusion included: not high income country, having a disease or role-related to those people with a specific health condition (for example heart disease), aged <60 years, baseline not a general population, data not reported, a study reported in more than one paper, published before 2008, and loneliness score were divided into quartile and quintiles and used as cut-offs.

5-According to author statement". There was a very high degree of heterogeneity" Please elaborate in discussion why there is high heterogeneity observed?

Our response:

Thank you for your comment. We have added the following statement:

“…likely due to the differences in the types of measurement tools used, different methods of obtaining data (online, face to face, postal questionnaire), differences in response rates as well as gender..”

6-Please provide rationale of meta-analysis results and explain in discussion

Our response:

Thank you for your comment, we have changed the statement in our discussion from:

Approximately 1 in 4 older adults over 60 experience loneliness at least sometimes. . .

TO

Our pooled estimate of loneliness prevalence (28.5%, 95%CI: 23.9% to 33.2%) suggests that approximately 1 in 4 older adults over 60 experience some degree of loneliness at least some of the time. Our a-priori subgroup analyses suggest that a smaller proportion of people (around 1 in 12) experience severe loneliness than moderate loneliness (approximately 1 in 4). There was no suggestion that loneliness is more common in people age over 75 than in those age 60-75.

7-It is recommended to draw the illustration of Geographical variation in prevalence of loneliness within and between countries mentioned inside the manuscript.

Our response:

Thank you for your comment, as this was not initially a focus in our review, there was insufficient data to merit the addition of a figure, however at the end of the paragraph, and we have acknowledged that:

Further research to better understand these geographical differences might support learning or understanding from countries with lower prevalence.

8-please provide rationale on high risk of bias in studies identified.

Our response:

Thank you for your suggestion. We have added the following statement:

The potential for bias was identified in 4 out of 9 domains, mainly due to not having enough participants to address the target population.

---

## [Decision Letter · Decision Letter 1]

12 Jul 2021

Prevalence of Loneliness amongst Older People in High-Income Countries: A Systematic Review and Meta-analysis

PONE-D-21-14347R1

Dear,

We’re pleased to inform you that your manuscript has been judged scientifically suitable for publication and will be formally accepted for publication once it meets all outstanding technical requirements.

Kind regards,

Muhammad Shahzad Aslam, Ph.D.,M.Phil., Pharm-D

Academic Editor

PLOS ONE

Additional Editor Comments (optional):

Reviewers' comments:

Reviewer's Responses to Questions

**Comments to the Author**

1. If the authors have adequately addressed your comments raised in a previous round of review and you feel that this manuscript is now acceptable for publication, you may indicate that here to bypass the “Comments to the Author” section, enter your conflict of interest statement in the “Confidential to Editor” section, and submit your "Accept" recommendation.

Reviewer #1: All comments have been addressed

Reviewer #2: All comments have been addressed

2. Is the manuscript technically sound, and do the data support the conclusions?

Reviewer #1: Yes

Reviewer #2: Yes

3. Has the statistical analysis been performed appropriately and rigorously? 

Reviewer #1: Yes

Reviewer #2: I Don't Know

4. Have the authors made all data underlying the findings in their manuscript fully available?

Reviewer #1: Yes

Reviewer #2: Yes

5. Is the manuscript presented in an intelligible fashion and written in standard English?

Reviewer #1: Yes

Reviewer #2: (No Response)

6. Review Comments to the Author

Reviewer #1: Dear. Author.

Many thanks for submitting your revised manuscript to this journal. Overall, the revised manuscript is improved and is relevant to the scope of this journal.

Reviewer #2: (No Response)

7. PLOS authors have the option to publish the peer review history of their article (what does this mean?). If published, this will include your full peer review and any attached files.

Reviewer #1: **Yes: **Dr. Yun Jin Kim, Ph.D

Reviewer #2: **Yes: **Saima Nisar

---

## [Editor Report · Acceptance letter]

14 Jul 2021

PONE-D-21-14347R1 

Prevalence of Loneliness amongst Older People in High-Income Countries: A Systematic Review and Meta-analysis 

Dear Dr. Kunonga:

I'm pleased to inform you that your manuscript has been deemed suitable for publication in PLOS ONE. Congratulations! Your manuscript is now with our production department. 

Kind regards, 

on behalf of

Dr. Muhammad Shahzad Aslam 

Academic Editor

PLOS ONE